# Association between early-pandemic food assistance use and subsequent food security trajectories among households in Washington State during the first three years of the COVID-19 pandemic

**James H. Buszkiewicz**[1], **Ashley S. Tseng**[2], **Jane Dai**[3], **Alan Ismach**[4], **Shawna Beese**[5,6], **Sarah M. Collier**[4,7], **Marie L. Spiker**[2,4‡], **Jennifer J. Otten**[4,7‡]*

1 Department of Epidemiology, Center for Social Epidemiology and Population Health, School of Public Health, University of Michigan, Ann Arbor, Michigan, United States of America, 2 Department of Epidemiology, School of Public Health, University of Washington, Seattle, Washington, United States of America, 3 Department of Health Systems and Population Health, School of Public Health, University of Washington, Seattle, Washington, United States of America, 4 Food Systems, Nutrition, and Health Program, School of Public Health, University of Washington, Seattle, Washington, United States of America, 5 College of Agricultural, Human, and Natural Resources Sciences, Washington State University, Pullman, Washington, United States of America, 6 College of Nursing, Washington State University, Spokane, Washington, United States of America, 7 Department of Environmental and Occupational Health Sciences, School of Public Health, University of Washington, Seattle, Washington, United States of America

‡ These authors are joint senior authors on this work.
* jotten@uw.edu

## Abstract

### Background

Research on COVID-19's impact on food insecurity has primarily relied on cross-sectional data or long recall periods, with limited investigations into longitudinal patterns or the role of food assistance.

### Methods

We analyzed longitudinal data from 703 respondents participating in at least three Washington State Food Security Survey waves between June 18, 2020, and January 7, 2023. We assessed food security using the United States Department of Agriculture's six-item module, categorizing respondents' trajectories as *persistently food secure*, *persistently food insecure*, or *experiencing one or more food insecurity transitions*. We categorized food assistance use as *never used*, *used before COVID-19 but not at baseline*, *did not use before COVID-19 but used at baseline*, or *always used*. We descriptively examined sociodemographic factors linked to each food security trajectory and food assistance use pattern. We assessed associations between food assistance use and food security trajectories using modified Poisson regression.

**Data availability statement:** Those interested in the data and variables used for the analyses presented in this manuscript can find complete data for the 703 respondents participating in three or more waves of Waves 1 through 4 of the Washington Food Security (WAFOOD) survey on the Dryad Data Repository using doi: https://doi.org/10.5061/dryad.02v6wwqdq.

**Funding:** This work was supported by the Washington State Department of Agriculture, the University of Washington Population Health Initiative, the University of Washington School of Public Health and Department of Environmental and Occupational Health, United of Way King County, the Paul G. Allen Foundation and other private philanthropy. The funders had no role in study design, data collection and analysis, decision to publish, or preparation of the manuscript.

**Competing interests:** The authors have declared that no competing interests exist.

## Results

We found that 20.2% of respondents were persistently food insecure, and 22.5% experienced one or more food insecurity transitions. Both patterns were more common among respondents who were aged 35 to 64, had a gender identity other than man or woman, were non-Hispanic Black, were single or divorced, had children, had some college education or less, reported $35,000 or less in household income, or were unemployed. In fully adjusted models, respondents who were newly using food assistance early in the COVID-19 pandemic had a higher probability of being persistently food insecure (marginal effect [ME] = 0.320, 95% CI = 0.204, 0.436) or experiencing one or more food insecurity transitions (ME = 0.216, 95% CI = 0.069, 0.363), than those who never used assistance.

## Conclusions

Our findings highlight the importance of examining food security trajectories and food assistance use patterns and implementing policies that help households new to food assistance programs navigate these systems.

## Introduction

During the first three years of the COVID-19 pandemic in the United States (U.S.), employment disruptions, business and school operational constraints, and food accessibility and price volatility created unique challenges for household food security (Fig 1) [1–4]. National unemployment jumped to 13.0% in the second quarter of 2020 compared to 3.8% in the first quarter of 2020 [5], and employment disruptions took many forms: some faced abrupt job loss during

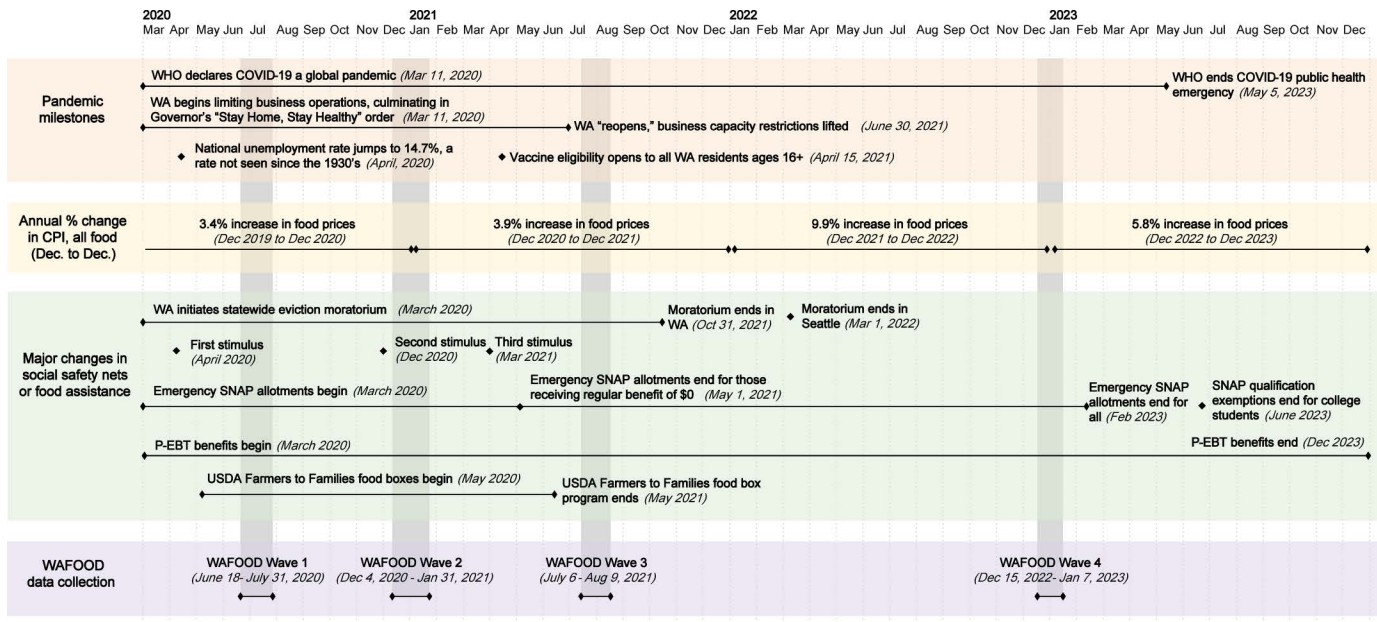

**Fig 1. Timeline of major COVID-19 pandemic milestones alongside events impacting household food security and WAFOOD survey data collection.** WHO = World Health Organization, WA = Washington State, CPI = Consumer Price Index, SNAP = Supplemental Nutrition Assistance Program, P-EBT=Pandemic EBT, USDA = United States Department of Agriculture, WAFOOD = Washington State Food Security Survey.

initial business closures, while others faced prolonged reduced job hours [1,4,6]. Families also faced additional disruptions, such as working and conducting school from home, which impacted routines and access to school meals [1,4,6]. Record food price inflation—brought on by supply chain disruptions, higher labor costs, and rising oil and goods prices due to international conflicts—further threatened household food security [1,4,6].

This confluence of pandemic-related threats to household food security meant that some households experienced food insecurity perhaps for the first time and may have been newly navigating an unfamiliar food assistance system [1]. Even households familiar with food assistance faced a system undergoing intermittent disruptions and evolution amidst unprecedented demand [1,4]. It is estimated that during 2020, at least 60 million Americans used food banks, food pantries, or other private food assistance programs, an increase of over 50% compared to 2019 [7]. Also, in 2020, the charitable food system distributed 44% more meals than the prior year [7]. In response to increased demand, government, private, and non-profit hunger relief organizations added new services; for example, the Pandemic EBT (P-EBT) program provided benefits to the families of children with limited onsite or childcare meals between March 2020 and December 2023 [8,9]. Some programs increased the benefit amounts of existing services; for example, federal boosts to the Supplemental Nutrition Assistance Program (SNAP) allowed eligible households to receive the maximum possible monthly benefit between March 2020 and February 2023 [9,10]. Other programs made services more accessible; for example, the Special Supplemental Nutrition Program for Women, Infants, and Children (WIC) programs conducted services virtually [11], and some food banks and pantries introduced delivery, curbside pick-up, mobile, or pop-up site services [12].

Much of the food security assessment during the early months of the COVID-19 pandemic in the U.S. relied on cross-sectional data or long recall periods (e.g., past 12 months) that could not capture recent or fluctuating food security changes [3]. This is a critical knowledge gap, as household food security status can change rapidly in response to social, economic, and policy contexts, which typified the months and years following initial COVID-19 disruptions [1,4]. potentially leading to underreporting of the food insecurity burden [13]. Before the COVID-19 pandemic, some studies with longitudinal data sought to map out life-course trajectories of food security [14–16]. These studies noted that while most participants remained food secure and very few remained food insecure, a substantial number of households experienced one or more periods of food insecurity, highlighting the importance of dynamic measures [14,15]. While some research groups have reported longitudinal data on household food security in the U.S. during the early years of the COVID-19 pandemic [17–25], less common were studies that characterized dynamic food security *trajectories* during the pandemic. One such study identified ten unique food security trajectories using biweekly surveys from April 2020 and March 2021 among a nationally representative sample of households. [3] They found that 64.7% of households remained food secure, 3.4% remained food insecure, and 35.3% experienced food insecurity at one or more time points. [3] They also observed that younger adults, women, and those who were from racially and ethnically minoritized backgrounds had the highest likelihood of remaining persistently food insecure. [3]

Although many of the acute economic impacts characterizing the early years of the COVID-19 pandemic have largely subsided, some impacts, like food price inflation, [26] have lingered. Food insecurity remains a serious public health concern in the U.S., with implications for health equity. [4] This study uses data from residents across Washington State (WA). Before the COVID-19 pandemic, WA food insecurity prevalence mirrored that of the U.S. at approximately 1 in 10, [1,27] with food insecurity highest among non-Hispanic Black and Hispanic or Latino households, low-income households, and households with children. [27] In WA, as in the U.S. more broadly, COVID-19 only served to exacerbate these inequities

through disproportionate impacts on rates of illness and death, unemployment, food access, and more [1,4,27,28].

Using data from the WA Food Security Survey (WAFOOD), a novel longitudinal survey of WA residents from 2020 to 2023, we sought to 1) describe food security trajectories and food assistance use patterns during the first three years of the COVID-19 pandemic, 2) describe distribution of these across sociodemographic characteristics, and 3) examine the associations between food assistance use patterns (specifically comparing use before COVID-19 and at baseline) and subsequent food security trajectories. This study adds to a growing body of literature prospectively examining food security trajectories and food assistance use patterns among U.S. households during COVID-19. To our knowledge, no studies have examined differences in food security trajectories between households with prior experiences using food assistance before the COVID-19 pandemic and households newly navigating the system. This work will inform ongoing policy efforts to reduce food insecurity and improve food assistance policies and programs.

## Methods

### Study design and population

We used data from the first four waves of WAFOOD, which covered a period marked by the initial COVID-19 business and school closures, food system disruptions, and rising inflation from June 18, 2020, to January 7, 2023 (Fig 1). The WAFOOD survey collected a rich array of data related to food security, economic security, food assistance use, barriers to food assistance use, diet, physical health, mental well-being, and sociodemographic characteristics. Researchers from the University of Washington and Washington State University developed the WAFOOD survey to monitor the impacts of the COVID-19 pandemic on food and economic security, food access, and prospects for recovery among WA residents. To design the WAFOOD survey, we used a combination of existing, validated survey tools and novel questions developed in collaboration with key partners. We sought input from personnel at local and state hunger relief and food distribution organizations and city and state public agencies regarding the survey design and its deployment, conducting over 40 hours of interviews in 2020 and approximately 10 hours for updates in preparation for each additional wave. Of the 72 questions include in the survey, 59 (82%) were adapted from existing tools that came from the U.S. Department of Agriculture's (USDA) Economic Research Service (ERS) Household Food Security Survey Module, [29,30] the Behavioral Risk Factor Surveillance System, [31] the National Health and Nutrition Examination Survey, [32] and surveys developed by our partners in the Nutrition and Obesity Policy Research and Evaluation Network, [33] the National Food Access and COVID Research Team, [34] and the Center for Nutrition and Health Impact, [35] with the remaining 13 questions created by the study team.

Respondents could access the WAFOOD survey, implemented online using REDCap, [36] using their computer, cellphone, or tablet. The survey was designed to take approximately 20 minutes. The WAFOOD survey sample was gathered using convenience sampling, with the survey advertised on Facebook and Instagram, and thus was not intended to be representative of the WA population. We made every effort to oversample lower-income households accessing local or state food assistance services by distributing the survey link across networks of over 400 hunger relief and food distribution organizations and public agency partners, who then shared the survey link via various electronic means with clients and community members. We fielded the Wave 1 survey from June 18 to July 31, 2020; the Wave 2 survey from December 4, 2020, to January 31, 2021; the Wave 3 survey from July 6 to August 9, 2021; and the Wave 4 survey from December 15, 2022, to January 7, 2023. We received 2,615 completed

responses in Wave 1, 3,501 in Wave 2, 3,074 in Wave 3, and 5,052 in Wave 4, totaling 14,242 completed responses. At the end of each survey, we asked respondents if they would be willing to participate in future survey waves. Each subsequent survey re-contacted this existing respondent pool while recruiting new respondents. Because of this, the WAFOOD survey has both cross-sectional and longitudinal components. As of the completion of the Wave 4 survey, the sample included 1,908 respondents participating in at least two waves, 785 respondents participating in three or more waves, and 223 respondents participating in four waves. Lastly, in each survey wave, we allowed respondents to provide an email address to enter a raffle to win one of 50 $50 gift cards to the grocery store of their choice.

Respondents needed to participate in three or more survey waves to be included in the present analysis and have complete sociodemographic, food security, and food assistance information. The final analytic sample was 703 respondents. A flow diagram with inclusion and exclusion criteria appears in S1 Fig. The University of Washington Institutional Research Board reviewed and approved the study and deemed it exempt from federal human subjects regulations. All survey questions included a *prefer not to answer* response option.

## Sociodemographic characteristics

Based on *a priori* hypotheses and prior associations in the literature, we examined several baseline sociodemographic factors linked to changes in food security. [3,18,25,37] These factors included age (18 to 34, 35 to 64, and ≥65 years), gender (women, men, or a gender identity other than man or woman), race and ethnicity (Hispanic, non-Hispanic Black, non-Hispanic Asian, non-Hispanic White, or another race and ethnicity), marital status (single or divorced, member of an unmarried couple, or married), the household has children (yes or no), educational attainment (some college or less or college degree or more), total annual household income (<$35,000, $35,000 to <$75,000, or ≥$75,000), and employment status (employed, unemployed, or not in the labor force).

We also examined differences based on urbanicity: whether the respondent lived in an urban or rural area based on the USDA ERS rural-urban commuting areas (RUCA) codes. [34] RUCA codes range from 1 *metropolitan area core* to 10 *rural areas*. Respondents were assigned a RUCA code based on their self-reported household ZIP code. We coded respondents as living in an urban area if their RUCA codes fell between 1 and 3. We coded respondents as living in a rural area if their RUCA codes fell between 4 and 10.

## Food security trajectories

We assessed food security using the validated U.S. Household Food Security Survey Module designed by the USDA ERS. [29,30] We fielded the 6-item U.S. Household Food Security Survey Module scale in Waves 1 and 2 and the 18-item scale in Waves 3 and 4. For consistency in this analysis, we report results based on the six items common to both scales. In accordance with the USDA ERS's standard scoring practices, [27,28] each respondent received a raw score based on their answers, which we used to assign food security categories. Scores of 0 or 1 indicated high or marginal food security, scores of 2–4 indicated *low food security*, and scores of 5 or 6 indicated *very low food security*. We dichotomized scores into food secure if respondents had high or marginal food security and food insecure if respondents had low or very low food security. In Wave 1 (fielded June 18 to July 31, 2020), respondents were asked to recall their food security *Since COVID-19*, defined as March 15, 2020. In Waves 2–4, we asked respondents to recall their food security in the past 30 days.

We then categorized respondents into three distinct food security trajectories based on patterns observed across all survey waves (three or four waves total, depending on the

respondent). We categorized respondents who were consistently food secure across all waves as *persistently food secure*. We categorized respondents who were consistently food insecure across all waves as *persistently food insecure*. We categorized respondents as *experienced one or more food insecurity transitions* if they were food secure at their baseline wave but experienced food insecurity at later waves; if they experienced food insecurity at their baseline wave but were food secure at later waves; or if they experienced more than one transition to or from food insecurity. There were instances of response gaps in survey waves (e.g., if an individual responded to Waves 1, 2, and 4 of the survey), for which we assumed no food security status changes during those unobserved gaps, recognizing this could underestimate changes in food security status among these respondents.

## Food assistance use before COVID-19 and at baseline

In all survey waves, we asked respondents whether they used one or more of several specific local, state, and federal food assistance programs. In Wave 1, we asked respondents if they or someone in their household used food assistance before COVID-19 (before March 15, 2020) (*yes* or *no*). Next, we asked respondents to select from a list of food assistance programs they or their family used, including SNAP, also known as Food Stamps, Electronic Benefits Transfer (EBT), or Basic Food (in WA); WIC; School Meal Programs (lunch or breakfast); summer meals from school; pick up at food banks or food pantries; grocery voucher or cash card provided from by city agency; mobile food banks delivery or pop-up sites near worksite; and other programs such as the Commodity Supplemental Food Program or Meals on Wheels. We then asked respondents when they had used each of these food assistance programs (*during the 12 months before COVID-19 [March 15, 2020], since COVID-19 [March 15, 2020]*, *never*, or *N/A*). At Wave 2, we slightly modified these response options (*during 2019 [before COVID-19], currently using*, *never*, or *N/A*). In addition, we asked respondents whether they used P-EBT (*yes*, *no*, or *I don't know what P-EBT is*). At Wave 3, regarding P-EBT, we asked respondents first if they had heard of P-EBT (*yes*, *no*, or *prefer not to answer*). We then asked if they used P-EBT (*yes, we have started receiving it*; *no, we are still waiting to receive it*; *no, we were denied*; or *no, I do not need P-EBT*). In addition, we also asked respondents in Wave 3 if they used preschool or childcare meal programs (child and adult care food program [CACFP]). In Wave 4, respondents were asked, *Which of the following food assistance programs has your household used in the past 30 days?* to which we consolidated all previously described food assistance programs into one check-all-that-apply list.

Based on food assistance usage patterns, we aggregated respondents into eight food assistance use categories (none; retail only, including SNAP, WIC, grocery vouchers, and P-EBT; school only, including school meals and summer meals; emergency only, including food banks/pantries, mobile boxes, community programs, and other; retail and school; retail and emergency; school and emergency; and all: retail, school, and emergency) (S1 Table). Due to small sample sizes within each category, we grouped respondents who used any food assistance program into one broad category, operationalizing food assistance use as a binary variable (any vs. none).

We then created a variable for food assistance use patterns that captured any transitions in food assistance program use before COVID-19 (March 15, 2020) and at each respondent's baseline wave, either Wave 1 (June 18 to July 31, 2020) or Wave 2 (December 4, 2020, to January 31, 2021). We categorized respondents into one of four categories (*never used*, *used before COVID-19 but not at baseline*, *did not use before COVID-19 but used at baseline*, or *always used*). *Never used* meant that a respondent neither used food assistance before COVID-19 nor at baseline. *Always used* meant that a respondent used food assistance before COVID-19 and at baseline.

## Statistical analysis

We calculated frequencies and percentages for food security trajectories, food assistance use patterns, sociodemographic characteristics, and urbanicity by baseline wave. Each respondent's baseline wave was defined as their first observed wave, either Wave 1 or Wave 2. Next, we calculated descriptive statistics to examine how food security trajectories and food assistance use patterns varied by demographic characteristics, socioeconomic factors, and urbanicity. We also used the ggplot2 package in R to generate alluvial plots to visualize food insecurity wave transitions by baseline wave and food assistance use patterns before COVID-19 and at baseline. We then fit a series of modified Poisson regression models [38] to estimate the predicted probability of being 1) *persistently food insecure* or 2) *experiencing one or more food insecurity transitions* relative to being *persistently food secure* associated with each food assistance use pattern: *never used, used before COVID-19 but not at baseline*, *did not use before COVID-19 but used at baseline*, or *always used*. Model 1 was unadjusted. Model 2 adjusted for baseline demographic characteristics (age, gender, race and ethnicity, marital status, and any children in the household). Model 3 adjusted for Model 2 and baseline socioeconomic factors (educational attainment, total annual household income, and current employment status). Lastly, Model 4 adjusted for Model 3 and additionally adjusted for urbanicity. We used each model's estimates to calculate predicted probabilities (PP) and 95% confidence intervals (CI) for each food assistance use pattern. We also calculated the difference in predicted probabilities, or marginal effect (ME), of each outcome associated with food assistance 1) *used before COVID-19 but not at baseline*, 2) *did not use before COVID-19 but used at baseline*, or 3) *always used* compared to *never used*. For ease of interpretation, we graphed predicted probabilities for each outcome for the fully adjusted Model 4. We performed statistical analyses using R version 4.2.2 and Stata 18 using a significance level of 0.05.

## Sensitivity analysis

We tested the robustness of our findings regarding the inclusion of respondents with missing food insecurity data. First, we restricted our sample to only those respondents with contiguous waves of food security data (i.e., no gaps between waves) (n = 465). Second, we restricted our analysis to only those respondents with complete food security data (i.e., respondents were excluded who chose *prefer not to answer* across food security questions) across waves (n = 356). Finally, we qualitatively compared the baseline sociodemographic characteristics of respondents included in our analytic sample to those respondents included in the larger WAFOOD Wave 1 and 2 cross-sectional samples and to WA residents and households using 2020 decennial U.S. Census Bureau [39] data and U.S. Census Bureau American Community Survey 5-year estimates [40].

# Results

## Sample characteristics

At their baseline wave (Wave 1 or Wave 2), respondents included in our analytic sample were primarily 35 to 64 years of age (61.5%), women (83.6%), and non-Hispanic White (80.9%) (Table 1). About half (47.7%) were married, and more than one-third (38.3%) had children. Over half (58.0%) of respondents had a college degree or more, and 59.7% were employed. For household income, 36.0% reported incomes less than $35,000, 30.3% reported incomes of $35,000 to $74,999, and 33.7% reported incomes of $75,000 or more. Most lived in urban areas (87.9%). About a third of respondents' households (32.4%) were food insecure. A third (35.6%) of respondents reported using food assistance before COVID-19. Likewise, one-third

**Table 1. Baseline respondent characteristics, WAFOOD Waves 1 and 2 (2020-2021) (n = 703).**

| Characteristic | n (%) |
|---|---|
| Age (years) | |
| 18 to 34 | 125 (17.8) |
| 35 to 64 | 432 (61.5) |
| 65 and older | 146 (20.8) |
| Gender | |
| Women | 588 (83.6) |
| Men | 87 (12.4) |
| A gender identity other than man or woman | 28 (4.0) |
| Race and ethnicity | |
| Hispanic | 38 (5.4) |
| Non-Hispanic Black | 14 (2.0) |
| Non-Hispanic Asian | 38 (5.4) |
| Non-Hispanic White | 569 (80.9) |
| Another race and ethnicity | 44 (6.3) |
| Marital status | |
| Single or divorced | 301 (42.8) |
| Member of an unmarried couple | 67 (9.5) |
| Married | 335 (47.7) |
| Household has children | |
| One or more children | 269 (38.3) |
| No children | 434 (61.7) |
| Educational attainment | |
| Some college or less | 295 (42.0) |
| College degree or more | 408 (58.0) |
| Total annual household income | |
| Less than $35,000 | 253 (36.0) |
| $35,000 to $74,999 | 213 (30.3) |
| $75,000 or more | 237 (33.7) |
| Employment status | |
| Employed | 420 (59.7) |
| Unemployed | 21 (3.0) |
| Not in labor force | 262 (37.3) |
| Urbanicity | |
| Urban | 618 (87.9) |
| Rural | 85 (12.1) |
| Food insecure at baseline | 228 (32.4) |
| Used any food assistance before COVID-19 | 250 (35.6) |
| Used any food assistance any baseline | 259 (36.8) |
| Baseline wave (participants' first observation) | |
| Wave 1 (June 18, 2020 to July 31, 2020) | 440 (62.6) |
| Wave 2 (December 4, 2020 to January 31, 2021) | 263 (37.4) |

Note: "A gender identity other than man or woman" includes transgender, nonbinary, and self-described. "Another race and ethnicity" includes American Indian or Alaskan Native, Native Hawai'ian or Other Pacific Islander, and self-described. "Single or divorced" additionally includes those who are widowed and separated. "Not in labor force" includes homemakers, students, retirees, and those unable to work. Use of any food assistance program included SNAP (Supplemental Nutrition Assistance Program, Food Stamps, Basic Food, or EBT), WIC (Women, Infants, and Children), grocery vouchers or cash card (provided by the city, food bank, food pantry, or other source), Pandemic-EBT (for respondents whose baseline was Wave 2), school meal programs (breakfast, lunch), summer meals from schools for children, food banks, mobile boxes (mobile food bank, drive-through, food gives, or other pop-up sites), food delivery from community programs, and other programs (e.g., Commodity Supplemental Food Program, Meals on Wheels).

(36.8%) of respondents reported using food assistance at baseline. S2 and S3 Figs compare the food security and food assistance in the present analytic sample to that of the larger, cross-sectional WAFOOD survey samples by wave. In the larger, cross-sectional WAFOOD survey samples, food insecurity ranged from 30% at Wave 1 to 49% at Wave 4 (S2 Fig), while food assistance use ranged from 32% at Wave 1 to 55% at Wave 4 (S3 Fig). S4 Fig visually represents food security transitions by food assistance use pattern and baseline wave. These alluvial plots illustrate the greater burden of food insecurity and transitions into and out of food security observed among respondents who used food assistance before COVID-19, at their baseline wave, or both compared to respondents who never used food assistance.

## Distribution of food security trajectories by sociodemographic characteristics and urbanicity

Among respondents in our analytic sample, 57.3% were persistently food secure across three or more waves from June 18, 2020, to January 7, 2023, while 20.2% were persistently food insecure, and 22.5% experienced one or more food security transitions (Table 2). Among those experiencing one or more transitions, 7.4% transitioned to and stayed food insecure, 7.4% transitioned to and stayed food secure, and 7.7% had one or more transitions to and from food insecurity (S1 Table). Persistent food insecurity was highest among respondents who were 35 to 64 years of age (24.8%), had a gender identity other than man or woman (28.6%), were non-Hispanic Black (42.9%), were single or divorced (28.6%), had children (29.4%), had some college education or less (30.5%), had total annual household income less than $35,000 (35.2%), were unemployed (48.6%), or were living in urban (20.4%) areas. These patterns were broadly similar for experiencing one or more food insecurity transitions, except for marital status, where the proportion of respondents experiencing one or more food insecurity was highest among those who were a member of an unmarried couple (34.3%)

## Distribution of food assistance use patterns by sociodemographic characteristics and urbanicity

Approximately 51.4% of respondents never used food assistance, 11.8% used food assistance before COVID-19 but not at baseline, 13.1% did not use food assistance before COVID-19 but use it at baseline, and 23.8% always used food assistance (Table 3). The sociodemographic distribution of food assistance use patterns tended to follow that observed for food security trajectories. For example, always using food assistance was most common among respondents who were 35 to 64 years of age (27.6%), had a gender identity other than man or woman (39.3%), were non-Hispanic Black (50.0%), were a member of an unmarried couple (32.8%) or single or divorced (31.9%), had children (36.4%), had some college education or less (32.2%), had total annual household incomes less than $35,000 (39.5%), were unemployed (47.6%), and lived in a rural area (29.4%). These distributions generally held across the other food assistance use patterns except for new use (did not use before COVID-19 but used at baseline) by income, as this pattern was evenly distributed across income levels (less than $35,000: 13.0%; $35,000 to $74,999: 13.2%; $75,000 or more: 13.1%), and by urbanicity, as pattern was more common among respondents living in urban (13.4%) rather than rural (10.6%) areas.

## Association between early-pandemic food assistance use and food security trajectories

In our adjusted model (Model 4), the highest probability of being persistently food insecure versus persistently food secure was observed among respondents who did not use food assistance before COVID-19 but did use food assistance at baseline (PP = 0.441, 95% CI =

**Table 2. Distribution of food security trajectories by sample sociodemographic characteristics and urbanicity at baseline observation, WAFOOD 1-4 (2020-2023) (n = 703).**

| Characteristic | Persistently food secure | Persistently food insecure | Experienced one or more food insecurity transitions |
|---|---|---|---|
| | n (%) | | |
| Overall | 403 (57.3) | 142 (20.2) | 158 (22.5) |
| Age (years) | | | |
| 18 to 34 | 76 (60.8) | 22 (17.6) | 27 (21.6) |
| 35 to 64 | 222 (51.4) | 107 (24.8) | 103 (23.8) |
| 65 and older | 105 (71.9) | 13 (8.9) | 28 (19.2) |
| Gender | | | |
| Women | 335 (57.0) | 119 (20.2) | 134 (22.8) |
| Men | 61 (70.1) | 15 (17.2) | 11 (12.6) |
| A gender identity other than man or woman | 7 (25.0) | 8 (28.6) | 13 (46.4) |
| Race and ethnicity | | | |
| Hispanic | 17 (44.7) | 12 (31.6) | 9 (23.7) |
| Non-Hispanic Black | 4 (28.6) | 6 (42.9) | 4 (28.6) |
| Non-Hispanic Asian | 21 (55.3) | 8 (21.1) | 9 (23.7) |
| Non-Hispanic White | 345 (60.6) | 98 (17.2) | 126 (22.1) |
| Another race and ethnicity | 16 (36.4) | 18 (40.9) | 10 (22.7) |
| Marital status | | | |
| Single or divorced | 133 (44.2) | 86 (28.6) | 82 (27.2) |
| Member of an unmarried couple | 33 (49.3) | 11 (16.4) | 23 (34.3) |
| Married | 237 (70.7) | 45 (13.4) | 53 (15.8) |
| Household has children | | | |
| One or more children | 124 (46.1) | 79 (29.4) | 66 (24.5) |
| No children | 279 (64.3) | 63 (14.5) | 92 (21.2) |
| Educational attainment | | | |
| Some college or less | 110 (37.3) | 90 (30.5) | 95 (32.2) |
| College degree or more | 293 (71.8) | 52 (12.7) | 63 (15.4) |
| Total annual household income | | | |
| Less than $35,000 | 81 (32.0) | 89 (35.2) | 83 (32.8) |
| $35,000 to $74,999 | 121 (56.8) | 47 (22.1) | 45 (21.1) |
| $75,000 or more | 201 (84.8) | 6 (2.5) | 30 (12.7) |
| Employment status | | | |
| Employed | 256 (61.0) | 76 (18.1) | 88 (21.0) |
| Unemployed | 5 (23.8) | 6 (28.6) | 10 (47.6) |
| Not in labor force | 142 (54.2) | 60 (22.9) | 60 (22.9) |
| Urbanicity | | | |
| Urban | 352 (57.0) | 126 (20.4) | 140 (22.7) |
| Rural | 51 (60.0) | 16 (18.8) | 18 (21.2) |

Note: All percentages are row percentages, "A gender identity other than man or woman" includes transgender, nonbinary, and self-described persons. "Another race and ethnicity" includes American Indian or Alaskan Native, Native Hawai'ian or Other Pacific Islander, and self-described persons. "Single or divorced" additionally includes those who are widowed and separated. "Not in labor force" includes homemakers, students, retirees, and those unable to work.

0.340, 0.542) (Fig 2, S2 Table). There was a statistically significantly higher probability of being persistently food insecure versus being persistently food secure among respondents who used food assistance before COVID-19 but not at baseline (ME = 0.118, 95% CI = 0.024, 0.212), did not use before COVID-19 but used at baseline (ME = 0.320, 95% = 0.204, 0.436), and always

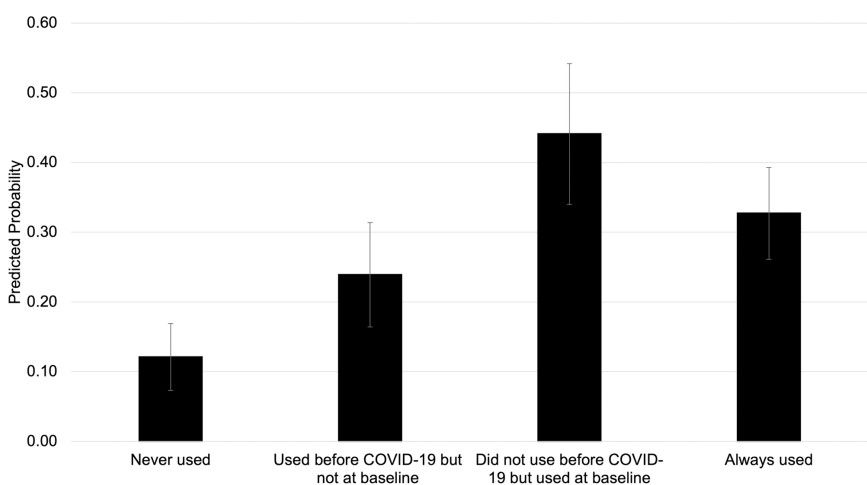

**Fig 2. Predicted probability of persistent food insecurity versus persistent food security associated with food assistance use before COVID-19 and at baseline, WAFOOD 1-4 (2020-2023) (n = 545).** Note: Sample sizes for each food security trajectory are "persistently food secure," n = 403 and "persistently food insecure," n = 142. Sample sizes for each food assistance use category are "neither used before COVID-19 nor at baseline," n = 361; "used before COVID-19 but not at baseline," n = 83; "did not use before COVID-19 but used at baseline," n = 92; and "used both before COVID-19 and at baseline," n = 167. A respondent's "baseline" measurement was determined as either Wave 1 or Wave 2, depending on whether the first WAFOOD survey they took was at Wave 1 or Wave 2. Food assistance use included the following programs: SNAP (Supplemental Nutrition Assistance Program, Food Stamps, Basic Food, or EBT), WIC (Women, Infants, and Children), grocery vouchers or cash cards (provided by the city, food bank, food pantry, or other source), Pandemic-EBT (for respondents whose baseline was wave 2), school meal programs (breakfast, lunch), summer meals from schools for children, food banks, mobile boxes (mobile food bank, drive-through, food gives, or other pop-up sites), food delivery from community programs, and other programs (e.g., Commodity Supplemental Food Program, Meals on Wheels). Estimates presented are from Model 4 and adjusted for age, gender, race and ethnicity, marital status, any children in the household, educational attainment, total annual household income, current employment status, and urbanicity at baseline.

used food assistance (ME = 0.206, 95% CI = 0.114, 0.298) compared to those who never used food assistance.

Similarly, in our fully adjusted model (Model 4), the highest probability of experiencing one or more food insecurity transitions versus being persistently food secure was observed among respondents who did not use food assistance before COVID-19 but did use food assistance at baseline (PP = 0.409, 95% CI = 0.279, 0.539) (Fig 3, S2 Table). There was a statistically significantly higher probability of experiencing one or more food insecurity transitions versus being persistently food secure among respondents used food assistance before COVID-19 but not at baseline (ME = 0.157, 95% CI = 0.036, 0.278), did not use before COVID-19 but used at baseline (ME = 0.216, 95% = 0.069, 0.363), and always used food assistance (ME = 0.143, 95% CI = 0.034, 0.252) compared to those who never used food assistance. These patterns were similar in the unadjusted models (Model 1), in models adjusting for demographic characteristics (Model 2), and in models adjusting for demographic and socioeconomic characteristics (Model 3). Estimates for main associations for all models can be found in S2 Table.

## Sensitivity analysis

Estimates derived from sensitivity analyses restricting our sample to those respondents 1) with contiguous waves of food security data and 2) with complete food security data across waves were similar in direction and magnitude compared with primary findings (S3 Table). Compared to respondents included in the larger cross-sectional WAFOOD Wave 1 and 2 samples,

**Table 3. Distribution of food assistance use patterns by sample sociodemographic characteristics and urbanicity at baseline observation, WAFOOD 1-4 (2020-2023) (n = 703).**

| | Never used | Used before COVID-19 but not at baseline | Did not use before COVID-19 but used at baseline | Always used |
|---|---|---|---|---|
| | n (%) | | | |
| Overall | 361 (51.4) | 83 (11.8) | 92 (13.1) | 167 (23.8) |
| Age (years) | | | | |
| 18 to 34 | 70 (56.0) | 13 (10.4) | 14 (11.2) | 28 (22.4) |
| 35 to 64 | 194 (44.9) | 55 (12.7) | 64 (14.8) | 119 (27.6) |
| 65 and older | 97 (66.4) | 15 (10.3) | 14 (9.6) | 20 (13.7) |
| Gender | | | | |
| Women | 292 (49.7) | 69 (11.7) | 83 (14.1) | 144 (24.5) |
| Men | 59 (67.8) | 8 (9.2) | 8 (9.2) | 12 (13.8) |
| A gender identity other than man or woman | 10 (35.7) | —ᵃ | —ᵃ | 11 (39.3) |
| Race and ethnicity | | | | |
| Hispanic | 16 (42.1) | 5 (13.2) | 8 (21.1) | 9 (23.7) |
| Non-Hispanic Black | —ᵃ | —ᵃ | —ᵃ | 7 (50.0) |
| Non-Hispanic Asian | 18 (47.4) | 5 (13.2) | 8 (21.1) | 7 (18.4) |
| Non-Hispanic White | 307 (54.0) | 66 (11.6) | 69 (12.1) | 127 (22.3) |
| Another race and ethnicity | 16 (36.4) | 6 (13.6) | 5 (11.4) | 17 (38.6) |
| Marital status | | | | |
| Single or divorced | 121 (40.2) | 47 (15.6) | 37 (12.3) | 96 (31.9) |
| Member of an unmarried couple | 27 (40.3) | 12 (17.9) | 6 (9.0) | 22 (32.8) |
| Married | 213 (63.6) | 24 (7.2) | 49 (14.6) | 49 (14.6) |
| Household has children | | | | |
| One or more children | 81 (30.1) | 35 (13.0) | 55 (20.5) | 98 (36.4) |
| No children | 280 (64.5 | 48 (11.1) | 37 (8.5) | 69 (15.9) |
| Educational attainment | | | | |
| Some college or less | 100 (33.9) | 49 (16.6) | 51 (17.3) | 95 (32.2) |
| College degree or more | 261 (64.0) | 34 (8.3) | 41 (10.1) | 72 (17.7) |
| Total annual household income | | | | |
| Less than $35,000 | 64 (25.3) | 56 (22.1) | 33 (13.0) | 110 (39.5) |
| $35,000 to $74,999 | 114 (53.5) | 17 (8.0) | 28 (13.2) | 54 (25.4) |
| $75,000 or more | 183 (77.2) | 10 (4.2) | 31 (13.1) | 13 (5.5) |
| Employment status | | | | |
| Employed | 242 (57.6) | 40 (9.5) | 57 (13.6) | 81 (19.3) |
| Unemployed | 5 (23.8) | 3 (14.9) | 3 (14.3) | 10 (47.6) |
| Not in labor force | 114 (43.5) | 40 (15.3) | 32 (12.2) | 76 (29.0) |
| Urbanicity | | | | |
| Urban | 326 (52.8) | 67 (10.8) | 83 (13.4) | 142 (23.0) |
| Rural | 35 (41.2) | 16 (18.8) | 9 (10.6) | 25 (29.4) |

ᵃCell values suppressed or complementarily suppressed due to small sample sizes (n < 3)

Note: All percentages are row percentages. Food assistance programs include SNAP (Supplemental Nutrition Assistance Program, Food Stamps, Basic Food, or EBT), WIC (Women, Infants, and Children), grocery vouchers or cash cards (provided by the city, food bank, food pantry, or other source), Pandemic-EBT (for respondents whose baseline was Wave 2), school meal programs (breakfast, lunch), summer meals from schools for children, food banks, mobile boxes (mobile food bank, drive-through, food gives, or other pop-up sites), food delivery from community programs, and other programs (e.g., Commodity Supplemental Food Program, Meals on Wheels). "A gender identity other than man or woman" includes transgender, nonbinary, and self-described persons. "Another race and ethnicity" includes American Indian or Alaskan Native, Native Hawai'ian or Other Pacific Islander, and self-described persons. "Single or divorced" additionally includes those who are widowed and separated. "Not in labor force" includes homemakers, students, retirees, and those unable to work.

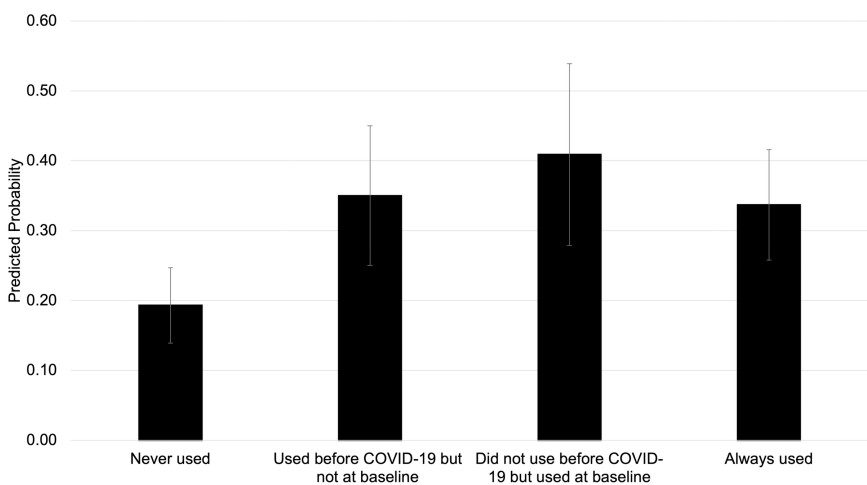

**Fig 3. Predicted probability of experiencing one or more food insecurity transitions versus persistent food security associated with food assistance use before COVID-19 and at baseline, WAFOOD 1-4 (2020-2023) (n = 561).** Note: Sample sizes for each food security trajectory are "persistently food secure," n = 403 and "people who experienced one or more food insecurity transitions," n = 158. Sample sizes for each food assistance use category are "neither used before COVID-19 nor at baseline," n = 361; "used before COVID-19 but not at baseline," n = 83; "did not use before COVID-19 but used at baseline," n = 92; and "used both before COVID-19 and at baseline," n = 167. A respondent's "baseline" measurement was determined as either Wave 1 or Wave 2, depending on whether the first WAFOOD survey they took was at Wave 1 or Wave 2. Food assistance use included the following programs: SNAP (Supplemental Nutrition Assistance Program, Food Stamps, Basic Food, or EBT), WIC (Women, Infants, and Children), grocery vouchers or cash cards (provided by the city, food bank, food pantry, or other source), Pandemic-EBT (for respondents whose baseline was wave 2), school meal programs (breakfast, lunch), summer meals from schools for children, food banks, mobile boxes (mobile food bank, drive-through, food gives, or other pop-up sites), food delivery from community programs, and other programs (e.g., Commodity Supplemental Food Program, Meals on Wheels). Estimates presented are from Model 4 and adjusted for age, gender, race and ethnicity, marital status, any children in the household, educational attainment, total annual household income, current employment status, and urbanicity at baseline.

a higher proportion of respondents in the longitudinal analytic sample presented here were 35 or older, non-Hispanic White, or had no children (S4 Table). A larger proportion of respondents in the longitudinal sample who entered in WAFOOD Wave 1 also had a college degree or more. However, it is important to note that sociodemographic patterns across samples were identical, and the cross-sectional survey samples have some item-specific non-response. Finally, compared to WA residents in 2020, [39] the longitudinal analytic sample presented here had a larger proportion of older, female, and non-Hispanic White adults and adults with low educational attainment and incomes below $75,000. Our sample also had more households with children than WA households in 2020.

## Discussion

We used longitudinal data from four waves (2020–2023) of the novel WAFOOD study, a convenience sample of Washington State adults oversampling lower-income households, to examine associations between food security trajectories and early pandemic transitions in food assistance use by sociodemographic factors. We found that the highest proportion of respondents who were persistently food-insecure or experienced one or more food insecurity transitions were those who were 35 to 64 years of age, non-Hispanic Black, single or divorced, or unemployed, and had children, another gender identity than man or woman, some college education or less, or a household income of $35,000 or less. This pattern was similar among those respondents who used food assistance before and during COVID-19. Importantly, we

found that relative to those respondents who never used food assistance before or during COVID-19, all other food assistance use patterns had a higher predicted probability of living in a persistently food insecure household or experiencing one or more food insecurity transitions than living in a persistently food secure household. However, the predicted probability of being persistently food insecure versus persistently food secure was greatest among those respondents who started using food assistance during the COVID-19 pandemic. Together, these findings reinforce known sociodemographic patterns of food security and food assistance use, highlight the importance of policies that help households new to food assistance programs more effectively and efficiently navigate these systems, and provide targets for programs aimed at improving health equity.

As with other local and state sites across the U.S., we observed that the burden of food insecurity and the demand for food assistance increased during the early months and years of the COVID-19 pandemic. [2] The pandemic only served to worsen existing sociodemographic disparities, [27,28,41–44] with the highest food insecurity burden and food assistance use among non-Hispanic Black respondents, respondents of low socioeconomic status, and those with children. Our study builds upon prior work by demonstrating that these households experienced a higher burden of both persistent and transitional food insecurity during the COVID-19 pandemic compared to non-Hispanic White respondents, higher socioeconomic status households, and households without children [17,20,22,24,37,44].

That we found both persistent and transitional food insecurity to be more common in some sociodemographic groups but not others is reflective of safety net system strengths and gaps alongside the complex interplay of social, economic, and structural forces driving U.S. food insecurity. [45,46] In line with prior work, [47,48] households with children in our sample had a higher probability of being persistently food insecure or experiencing one or more food insecurity transitions. Before COVID-19, targeted policy efforts reduced child food insecurity to 13.6% in 2019, but economic shocks in 2020 caused a spike that was briefly alleviated to 12.5% by government (e.g., expanding WIC options, maximizing SNAP allotments) and pandemic-era programs (e.g., P-EBT, Child Tax Credit); however, the expiration of some waivers and of pandemic-era programs led to a sharp increase to 17.3% in 2022. [45,46] Likewise, we observed that food insecurity transitions and persistent food insecurity were more common among non-Hispanic Black and Hispanic respondents and respondents of low socioeconomic status. During COVID-19, these populations experienced higher unemployment [49–51] and higher case rates and deaths [51] related to COVID-19 [52] compared to their non-Hispanic White and high socioeconomic counterparts, continuing pre-pandemic trends. [45,46,49,50] Conversely, adults over the age of 65 in this analysis experienced lower food insecurity transitions and were less likely to be persistently food insecure, a finding that has been reflected in other work, which may be due to the higher availability of food assistance programs for older rather than younger adults. [53] Our work, in the context of a rich body of work examining U.S. food insecurity [54,55] highlights opportunities to build a more resilient food system and hunger relief program network through partnerships between researchers, hunger relief organizations, government organizations, and policymakers to identify vulnerable populations and develop targeted policies and interventions.

Outside of the context of the COVID-19 pandemic, connections between food assistance use and food security status have been well-established, especially for SNAP. [56,57] For example, although participation in SNAP is more common among those experiencing food insecurity than those who are food secure, SNAP has been shown to alleviate food insecurity among participants. [15,58–61] However, to our knowledge, our study is the first to examine early-pandemic food assistance use (before the pandemic and at participants' baseline observation during the pandemic). Like prior work, we observed that food assistance use was

associated with food insecurity. [15,58–61] However, we observed the strongest relationship with food insecurity, particularly persistent food insecurity, among respondents who started using food assistance at their baseline observation, which would have been during the first year of the pandemic. This relationship may indicate the depth of the economic disruption experienced by many WA households, the scale and speed of changes occurring within the food safety net system, or challenges in newly navigating food assistance programs efficiently and effectively during the COVID-19 pandemic. As local, state, federal, and charitable food assistance programs, such as SNAP, [62] continue to modernize in ways that reduce participant burden (e.g., through investments in automation or online and mobile technologies) and improve outreach, food assistance programs should monitor new entrants to food assistance programs to identify and resolve continued barriers to use.

Our work also has important implications for public health and health equity. Food insecurity, even transient food insecurity, has been linked to many adverse health outcomes, including poor diet and poor mental well-being. [4,62–68] Food insecurity can have both direct—through prolonged poor diet quality—and indirect impacts on cardiometabolic health, including through physiologic dysregulation related to chronic exposure to psychosocial stressors, causing inflammation and damage to the immune system, placing individuals at greater risk of developing chronic diseases. [4] One notable study by Kong et al. examined the impact of food insecurity on mental health trajectories during COVID-19 using the Understanding Coronavirus in America survey. [18] They found that symptoms of stress, anxiety, and depression remained stable or improved among respondents living in food-secure households but that mental health declined among respondents living in food-insecure households, with the mental health gap between food-secure and insecure households widening throughout the pandemic. [18] However, to the authors' knowledge, the impact of transient versus persistent household food insecurity on adult health, particularly during the COVID-19 pandemic, has not been well-studied. [69,70] Future studies should examine the impact of persistent and transient food security trajectories on adult physical and mental health outcomes and whether food assistance program participation lowers the risk of poor health outcomes.

Our study had several notable strengths. First, our study used novel WAFOOD study data, which combined widely accepted measures of food and economic security with questions about food assistance use adapted to the local context of WA and the temporal context of the COVID-19 pandemic. Second, we were able to follow respondents longitudinally to assess food security trajectories, a limitation of prior cross-sectional COVID-19 food security surveys. [17–25] Third, to our knowledge, this study represents the first evaluation of the impact of new entry into food assistance use on subsequent food security trajectories. Fourth, rather than using a food security screener, we administered all food security scale questions to all respondents, not just those below 185% Federal Poverty Level who respond affirmatively to the first three questions on the 18-item scale, as done in the Current Population Survey Food Security Supplement. Administering all survey items to participants at all income levels may have been particularly important during COVID-19, as employment disruptions may have created new food insecurity risks among respondents at higher income levels.

Despite our study's many strengths, there were several important limitations. First, WAFOOD respondents were not representative of all WA residents. Participants were recruited via social media and through non-profit hunger relief organizations and public agency networks, and the survey was fielded entirely online, which may have missed households without internet access or computers, cell phones, or tablets. However, it was this sampling strategy that enabled us to recruit households most burdened by food insecurity and map and describe food security trajectories. Second, due to potential respondent recall issues, we did not assess household food security status before the COVID-19 pandemic and could not compare these estimates before

and during the COVID-19 pandemic. Third, although our evaluation of food security occurs over a longer period than prior studies assessing food security during the pandemic, we only assessed food security at four time points and at uneven but seasonally matched time intervals, limiting our ability to characterize food security trajectories at greater detail. Fourth, we used different USDA food security scales over time. We used the 6-item scale in early waves and the 18-item scale in later waves, limiting our assessment of longitudinal trajectories to the six-item scale. However, the six-item scale is more sensitive to capturing food security than one- to three-item questionnaires, such as those used by the U.S. Census Bureau's Household Pulse Survey [71] and the Understanding Coronavirus in America survey. [72] Finally, small sample sizes impacted our analyses in several key ways: (a) though other researchers have characterized up to 10 classes of food insecurity trajectories, [3] we were unable to examine food security trajectories across more than three categories; (b) although we assessed food assistance program use in great detail, ultimately, we were unable to assess the impact of each of these programs individually or by type (e.g., government assistance, such as SNAP); and (c) we were unable to examine whether the relationship between early-pandemic food assistance use and subsequent food security trajectories varied across sociodemographic factors.

The present study found new early pandemic food assistance use was linked to a greater likelihood of experiencing food insecurity during the first three years of the COVID-19 pandemic among a convenience sample of WA adults. While 57.3% of respondents remained persistently food-secure during three or more survey waves, 20.2% remained persistently food-insecure, and 22.5% experienced one or more food security transitions. Importantly, we found that although any food assistance use was associated with food insecurity, those respondents who started using food assistance programs during the first year of the COVID-19 pandemic had the greatest odds of persistent food insecurity or experiencing one or more food security transitions. Our findings highlight the importance of examining food security trajectories and food assistance use patterns and implementing policies that help households new to food assistance programs navigate these systems.

## Supporting information

**S1 Fig. Analytic sample inclusion and exclusion flow diagram, WAFOOD 1-4 (2020-2023).**
(PDF)

**S2 Fig. Food insecurity for the WAFOOD cross-sectional samples (A) and the longitudinal sample of respondents participating in three or more survey waves (B), WAFOOD 1-4 (2020-2023).**
(PDF)

**S3 Fig. Food assistance use for WAFOOD cross-sectional samples (A) and the longitudinal sample of respondents participating in three or more survey waves (B), WAFOOD 1-4 (2020-2023).**
(PDF)

**S4 Fig. Food security trajectories by food assistance use before COVID-19 and baseline food assistance use at baseline and baseline WAFOOD survey wave, WAFOOD 1-4 (2020-2023).**
(PDF)

**S1 Table. Detailed food assistance use categories at baseline observation and food security trajectories, WAFOOD 1-4 (2020-2023) (N = 703).**
(PDF)

**S2 Table. Associations between food assistance use before COVID-19 and at baseline and food security trajectories, WAFOOD 1-4 (2020-2023) (N = 703).**
(PDF)

**S3 Table. Sensitivity analyses examining the associations between food assistance use before COVID-19 and at baseline and food security trajectory outcomes, WAFOOD 1-4 (2020-2023).**
(PDF)

**S4 Table. Baseline characteristics of the WAFOOD longitudinal analytic and cross-sectional survey samples based on wave of entry, WAFOOD 1 and 2 (2020-2021), compared to Washington State residents (2020).**
(PDF)

## Acknowledgments

The WAFOOD team wishes to thank the Washington residents who shared their time and information in completing this survey, the Washington State Department of Agriculture, the University of Washington (UW) Population Health Initiative, the UW School of Public Health and Department of Environmental and Occupational Health (DEOHS), the Washington State Department of Agriculture, the Paul G. Allen Family Foundation, and other private philanthropy for their generous support in funding WAFOOD survey waves, and United Way of King County for supporting survey translations. We thank the numerous community partners and stakeholders who helped shape this project. Among those are the Washington State Department of Health, Washington State Department of Agriculture, the Washington State Anti-Hunger & Nutrition Coalition, Washington State SNAP-Ed, King County Local Food Initiative, Northwest Harvest, Washington State University (WSU) Extension, United Way of Washington, and numerous food banks, food pantries, charitable organizations, community organizations, county health departments, and local health jurisdictions. The Nutrition and Obesity Policy Research and Evaluation Network (NOPREN) and the ad-hoc COVID-19 Food Security Surveys subgroup shared valuable insights and surveys relevant to this project. We also wish to thank past WAFOOD team members for their contributions to the project. Finally, we thank Yoonseo Mok and Jihyoun Jeon at the University of Michigan School of Public Health for developing and sharing the ggplot R code used to create our alluvial plots.

## Author contributions

**Conceptualization:** James H. Buszkiewicz, Marie L. Spiker, Jennifer J. Otten.

**Data curation:** Ashley S. Tseng, Jane Dai.

**Formal analysis:** James H. Buszkiewicz, Ashley S. Tseng, Jane Dai.

**Funding acquisition:** Marie L. Spiker, Jennifer J. Otten.

**Investigation:** James H. Buszkiewicz, Ashley S. Tseng, Jane Dai, Marie L. Spiker, Jennifer J. Otten, Shawna Beese, Alan Ismach.

**Methodology:** James H. Buszkiewicz, Marie L. Spiker, Jennifer J. Otten.

**Project administration:** Marie L. Spiker, Jennifer J. Otten.

**Resources:** Marie L. Spiker, Jennifer J. Otten.

**Supervision:** Marie L. Spiker, Jennifer J. Otten.

**Visualization:** Ashley S. Tseng, Jane Dai, James H. Buszkiewicz, Marie L. Spiker, Alan Ismach.

**Writing – original draft:** James H. Buszkiewicz, Marie L. Spiker.

**Writing – review & editing:** James H. Buszkiewicz, Ashley S. Tseng, Jane Dai, Alan Ismach, Shawna Beese, Sarah M. Collier, Marie L. Spiker, Jennifer J. Otten.

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
