## [Decision Letter · Decision Letter 0]

11 Dec 2024

PONE-D-24-22286Association between early-pandemic food assistance use and subsequent food insecurity trajectories among households in Washington State during the first three years of the COVID-19 pandemicPLOS ONE

Dear Dr. Buszkiewicz,

Thank you for submitting your manuscript to PLOS ONE. After careful consideration, we feel that it has merit but does not fully meet PLOS ONE’s publication criteria as it currently stands. Therefore, we invite you to submit a revised version of the manuscript that addresses the points raised during the review process.

Please also note that you may be receiving another review. There was one review outstanding but the reviewer has gone beyond the deadline. If they should send in their review I will forward to you.

We look forward to receiving your revised manuscript.

Kind regards,

Karen M Davison, PhD

Academic Editor

PLOS ONE

Journal Requirements:

3. In the online submission form, you indicated that data cannot be shared publicly because of privacy concerns. However, data are available from the Washington Food Security Survey team upon request and completion of a data use agreement. 

Reviewers' comments:

Reviewer's Responses to Questions

**Comments to the Author**

1. Is the manuscript technically sound, and do the data support the conclusions?

Reviewer #1: Yes

Reviewer #2: Yes

2. Has the statistical analysis been performed appropriately and rigorously? 

Reviewer #1: No

Reviewer #2: Yes

3. Have the authors made all data underlying the findings in their manuscript fully available?

Reviewer #1: Yes

Reviewer #2: Yes

4. Is the manuscript presented in an intelligible fashion and written in standard English?

Reviewer #1: Yes

Reviewer #2: Yes

5. Review Comments to the Author

Reviewer #1: Thank you for the opportunity to review this manuscript. I really enjoyed reading it and I think it will make an excellent contribution to the literature. The manuscript is well-written, concise, and easy to follow. I just have a few questions and comments:

Why were longitudinal regression analyses not used (e.g., generalized estimating equations)? The analyses should account for repeated measures within individuals.

Thank you for including Supplemental Figures 2 and 3 and comparing the longitudinal sample to the larger cross-sectional sample in terms of food insecurity and food assistance use. Were the two samples similar in terms of demographic characteristics? It might be helpful to readers to include one or two sentences clarifying this.

I understand the value in presenting the different models that were built, but in the Results section, there’s a lot of focus on the unadjusted models and I’m not sure why. It seems like there should be more of a focus (or a complete focus) on the adjusted models, particularly the fully adjusted model. Can the authors clarify why they present and interpret results from the unadjusted models when those coefficients could be biased due to confounding?

How many questions was the WAFOOD, and which WAFOOD questions came from validated preexisting surveys (e.g., USDA Household Food Security Survey Module, BRFSS, NHANES) and which ones were new? How long did it take participants to complete the survey?

I appreciate the authors acknowledging the convenience sample as a limitation throughout the manuscript. It might be useful to readers to expand on this more in the Discussion- for example, if this sample was representative (i.e., not mostly urban, college educated white women), how might the results change? Would the associations be even stronger (especially given that the sample may have missed households without internet access or computers, cell phones, or tablets, which could be a proxy for low-income/high food insecurity)? It might also be helpful to specifically state in the limitations that this sample was mostly urban white women (e.g., modifying this sentence to something like: First, WAFOOD respondents were not representative of all WA residents; they were mostly urban, white women, while the greater WA state population consists of a much larger population of _____.”).

I think the Discussion section would benefit from a deeper discussion on why certain demographic characteristics (e.g., age [35 to 64 years of age], race/ethnicity [non-Hispanic Black], marital status [being single or divorced], gender identity) may have had the highest proportion of food insecurity or experiencing a food insecurity transition, and what are the policy implications of these results. For example, maybe higher age was protective because there were special programs for seniors that protected against food insecurity during the pandemic, such as Meals on Wheels, free grocery delivery for seniors, or just the fact that many seniors live in assisted living homes or live with caretakers who can cook and shop for them (e.g., other family members). Perhaps food assistance programs should therefore focus more on assisting younger populations. You could also cite that other studies have found that higher age may have been protective against food insecurity during the pandemic (e.g., Chapman et al. found that starting at age 60, higher age became protective against food insufficiency during the first year of the COVID-19 pandemic, such that participants ages 60–69, 70–79, and 80+ had a 13%, 49%, and 48% lower odds of food insufficiency, respectively.). (https://www.tandfonline.com/doi/full/10.1080/19320248.2023.2202618)

“National unemployment jumped to 14.7% in April 2020” - jumped from what to 14.7%? Was this a large jump? I’m assuming so, but it would be helpful to clarify.

“It is estimated that during 2020, at least 60 million Americans used food banks, food pantries, or other private food assistance programs…” compared to how many normally? Is 60 million a large jump? Again, I assume so, but might be helpful to clarify.

Figure 1 is blurry- can the authors improve the quality so it is more readable?

“some impacts, like food price inflation, have lingered.” I think this may need a citation- perhaps: https://www.ers.usda.gov/data-products/ag-and-food-statistics-charting-the-essentials/food-prices-and-spending/?topicId=1afac93a-444e-4e05-99f3-53217721a8be (“US food prices rose by 25 percent from 2019 to 2023”).

“This study uses data from residents across Washington State (WA), which was

the site of the first US COVID-19 death in March 2020.” – why is this important? Are the authors implying that WA State may have been hit more intensely than other states since they were the site of the first death and therefore dealt with the consequences of COVID-19 for longer?

“Each respondent received a raw score based on their answers, which we used to assign food security categories. Scores of 0 or 1 indicated high or marginal food security, scores of 2-4 indicated low food security, and scores of 5 or 6 indicated very low food security. We dichotomized scores into food secure if respondents had high or marginal food security and food insecure if respondents had low or very low food security.” – I think you can state that this is in accordance with the USDA ERS’s standard scoring practices and then cite this (page 4, where it says “For some reporting purposes, the food security status of households with raw score 0-1 is

described as food secure and the two categories “low food security” and “very low food security” in combination are referred to as food insecure.”): https://www.ers.usda.gov/media/8282/short2012.pdf. I think this might be your Reference #28. Just a suggestion so that readers don’t think you’re changing the survey’s scoring practices.

Reviewer #2: This is a wonderfully put-together research article that highlights a significant public health issue. The results were displayed in an easy-to-read, approachable manner.

No changes seem to be required.

6. PLOS authors have the option to publish the peer review history of their article (what does this mean? ). If published, this will include your full peer review and any attached files.

**Do you want your identity to be public for this peer review?** For information about this choice, including consent withdrawal, please see our Privacy Policy .

Reviewer #1: No

Reviewer #2: **Yes: ** Zainab Syyeda Rahmat

---

## [Author Response · Author response to Decision Letter 1]

9 Feb 2025

Reviewer #1

Thank you for the opportunity to review this manuscript. I really enjoyed reading it and I think it will make an excellent contribution to the literature. The manuscript is well-written, concise, and easy to follow. I just have a few questions and comments:

We thank Reviewer #1 for their kind words and for their time in providing a thorough review and thoughtful feedback on our manuscript.

Why were longitudinal regression analyses not used (e.g., generalized estimating equations)? The analyses should account for repeated measures within individuals.

The reviewer makes an excellent point, and we completely agree. Our original analysis, conducted in R, did not allow us to account for repeated measures using ‘mlogit.’ We now conduct the analyses using Stata 18. We have further opted to use generalized estimating equations using the family Poisson, log link, and an independent correlation matrix to account for repeated observations nested within respondents, along with robust standard errors (a modified Poisson regression model). We also use the Stata ‘margins’ post-estimation command to calculate the predicted probabilities of persistently food insecure or experiencing one or more food insecurity transitions relative to being persistently food secure over each food assistance use pattern. We also calculate the difference in these predicted probabilities or the ‘marginal effect’ for each food assistance use pattern relative to having never used food assistance (before COVID-19 or at baseline).

We describe this new analysis on page 9, paragraph 1 of the ‘Statistical analysis’ sub-section of the ‘Methods’ section. We describe the results of these new analyses on page 16, paragraphs 2 and 3 of the ‘Association between early-pandemic food assistance use and food security trajectories’ sub-section of the ‘Results’ section. Full main association estimates are presented in Supplemental Table 2 while a visual representation of results from the fully adjusted Model 4 are presented in Figures 2 and 3.

Thank you for including Supplemental Figures 2 and 3 and comparing the longitudinal sample to the larger cross-sectional sample in terms of food insecurity and food assistance use. Were the two samples similar in terms of demographic characteristics? It might be helpful to readers to include one or two sentences clarifying this.

We agree with the reviewer that including a table comparing the sociodemographic characteristics of our sample to the larger WAFOOD Wave 1 and 2 cross-sectional samples would be useful context for readers in addition to the information we have already provided regarding food insecurity and food assistance use. In Supplemental Table 4 we now include descriptive statistics of our longitudinal sample, based on WAFOOD wave of entry (1 or 2) and the WAFOOD Wave 1 and 2 cross-sectional samples as well as to Washington State residents and households in 2020 using US Census bureau decennial census data and US Census American Community Survey 5-year estimates. A comparison to the aggregate longitudinal analytic sample can be achieved by comparing these descriptive statistics to Table 1.

We provide a description of this new analysis in paragraph 1 on page 11 of the ‘Sensitivity analysis’ sub-section of the ‘Methods’ section:

“Finally, we qualitatively compared the baseline sociodemographic characteristics of respondents included in our analytic sample to those respondents included in the larger WAFOOD Wave 1 and 2 cross-sectional samples and to WA residents and households using 2020 decennial U.S. Census Bureau39 data and U.S. Census Bureau American Community Survey 5-year estimates.40.”

We describe the results on paragraph 1 on page 18 and 19 of the ‘Sensitivity analysis’ sub-section of the ‘Results’ section:

“Estimates derived from sensitivity analyses restricting our sample to those respondents 1) with contiguous waves of food security data and 2) with complete food security data across waves were similar in direction and magnitude compared with primary findings (Supplemental Table 3). Compared to respondents included in the larger cross-sectional WAFOOD Wave 1 and 2 samples, a higher proportion of respondents in the longitudinal analytic sample presented here were 35 or older, non-Hispanic White, or had no children (Supplemental Table 4). A larger proportion of respondents in the longitudinal sample who entered in WAFOOD Wave 1 also had a college degree or more. However, it is important to note that sociodemographic patterns across samples were identical and the cross-sectional survey samples have some item-specific non-response. Finally, compared to WA residents in 2020,39 the longitudinal analytic sample presented here had a larger proportion of older, female, and non-Hispanic White adults and adults with low educational attainment and incomes below $75,000. Our sample also had more households with children than WA households in 2020.”

I understand the value in presenting the different models that were built, but in the Results section, there’s a lot of focus on the unadjusted models and I’m not sure why. It seems like there should be more of a focus (or a complete focus) on the adjusted models, particularly the fully adjusted model. Can the authors clarify why they present and interpret results from the unadjusted models when those coefficients could be biased due to confounding?

We again completely agree with the reviewer. We have completely reorganized the results section to now focus solely on associations estimated from fully adjusted models. We describe the results of these new analyses on page 17, paragraphs 1 and 2 of the ‘Associations between early-pandemic food assistance use and food security trajectories’ sub-section of the ‘Results’ section and present them visually in Figures 2 and 3. We think the focus on fully adjusted Model 4 results and presenting the findings in figures makes the results and narrative clearer. Estimates for Models 1-3 along with full Model 4 estimates are preserved and presented in Supplemental Table 2.

In addition, to test the robustness of our findings, we now also include two additional sensitivity analyses that examine the impact of missing food security data on these estimates by further restricting our analytic sample to 1) those respondents with contiguous waves of food security data and 2) respondents with complete food security data across waves. We provide a description of this new analysis in paragraph 1 on page 10 and 11 of the ‘Sensitivity analysis’ sub-section of the ‘Methods’ section:

“We tested the robustness of our findings regarding the inclusion of respondents with missing food insecurity data. First, we restricted our sample to only those respondents with contiguous waves of food security data (i.e., no gaps between waves) (n = 465). Second, we restricted our analysis to only those respondents with complete food security data (i.e., respondents were excluded who chose “prefer not to answer” across food security questions) across waves (n = 356).”

We describe the results on paragraph 1 on page 18 of the ‘Sensitivity analysis’ sub-section of the ‘Results’ section:

“Estimates derived from sensitivity analyses restricting our sample to those respondents 1) with contiguous waves of food security data and 2) with complete food security data across waves were similar in direction and magnitude compared with primary findings (Supplemental Table 3).”

How many questions was the WAFOOD, and which WAFOOD questions came from validated preexisting surveys (e.g., USDA Household Food Security Survey Module, BRFSS, NHANES) and which ones were new? How long did it take participants to complete the survey?

59 of the 72 questions of the WAFOOD survey were taken from existing surveys, with the remaining questions generated anew by this study team. The survey was designed to take approximately 20 minutes.

We now include this information in paragraph 1 and 2 on page 5 in the ‘Study design and population’ sub-section of the ‘Methods’ section:

“Of the 72 questions included in the survey, 59 (82%) were adapted from existing tools that came from the US Department of Agriculture’s (USDA) Economic Research Service (ERS) Household Food Security Survey Module,29,30 the Behavioral Risk Factor Surveillance System,31 the National Health and Nutrition Examination Survey,32 and surveys developed by our partners in the Nutrition and Obesity Policy Research and Evaluation Network,33 the National Food Access and COVID Research Team,34 and the Center for Nutrition and Health Impact,35 with the remaining 13 questions created by the study team.”

“Respondents could access the WAFOOD survey, implemented online using REDCap,34 using their computer, cellphone, or tablet. The survey was designed to take approximately 20 minutes.”

I appreciate the authors acknowledging the convenience sample as a limitation throughout the manuscript. It might be useful to readers to expand on this more in the Discussion- for example, if this sample was representative (i.e., not mostly urban, college educated white women), how might the results change? Would the associations be even stronger (especially given that the sample may have missed households without internet access or computers, cell phones, or tablets, which could be a proxy for low-income/high food insecurity)? It might also be helpful to specifically state in the limitations that this sample was mostly urban white women (e.g., modifying this sentence to something like: First, WAFOOD respondents were not representative of all WA residents; they were mostly urban, white women, while the greater WA state population consists of a much larger population of _____.”).

We thank the reviewer for the suggestion. As mentioned above, in Supplemental Table 4, we now provide a direct comparison to 2020 Washington State residents and households as a whole. This comparison illustrates that compared to Washington State residents and households in 2020, our population had higher representation from older, female, non-Hispanic White, less educated adults, as well as higher representation from households with children and incomes below $75,000. While the reviewer’s suggestion that excluding households without internet access or electronic devices would like lead to an underestimation of ‘true’ population-level associations, we believe that given our respondent sample’s sociodemographic profile, estimates would likely be attenuated if the sample had been state representative. However, we do not wish to speculate on what we would have found had our sample been representative, rather we would like to focus on the strengths of our sample recruited in partnership with local and state hunger relief organizations.

We include the following text in paragraph 1 on page 19 of the ‘Sensitivity analysis’ sub-section of the ‘Results’ section:

“Finally, compared to WA residents in 2020,39 the longitudinal analytic sample presented here had a larger proportion of older, female, and non-Hispanic White adults and adults with low educational attainment and incomes below $75,000. Our sample also had more households with children than WA households in 2020.”

We also include the following text in paragraph 5 on page 23 of the ‘Discussion’ section:

“However, it was this sampling strategy that enabled us to recruit households most burdened by food insecurity and map and describe food security trajectories”

I think the Discussion section would benefit from a deeper discussion on why certain demographic characteristics (e.g., age [35 to 64 years of age], race/ethnicity [non-Hispanic Black], marital status [being single or divorced], gender identity) may have had the highest proportion of food insecurity or experiencing a food insecurity transition, and what are the policy implications of these results. For example, maybe higher age was protective because there were special programs for seniors that protected against food insecurity during the pandemic, such as Meals on Wheels, free grocery delivery for seniors, or just the fact that many seniors live in assisted living homes or live with caretakers who can cook and shop for them (e.g., other family members). Perhaps food assistance programs should therefore focus more on assisting younger populations. You could also cite that other studies have found that higher age may have been protective against food insecurity during the pandemic (e.g., Chapman et al. found that starting at age 60, higher age became protective against food insufficiency during the first year of the COVID-19 pandemic, such that participants ages 60–69, 70–79, and 80+ had a 13%, 49%, and 48% lower odds of food insufficiency, respectively.).

We completely agree with the reviewer and have added a new paragraph in the discussion section which attempts to highlight some of the reasons why certain sociodemographic groups may be more heavily impacted by persistent or transitory food insecurity and what the policy implications of these findings are. For the sake of brevity, we have chosen to only give a couple of examples rather than provide rationale and policy implications for all subpopulations examined in this manuscript. We have chosen to highlight racial and ethnic differences and age differences, thanks to the reviewer’s suggestion.

In paragraph 3 on page 20 and 21 in the Discussion section we have added the following text:

“That we found both persistent and transitional food insecurity to be more common in some sociodemographic groups but not others is reflective of safety net system strengths and gaps alongside the complex interplay of social, economic, and structural forces driving U.S. food insecurity.45,46 In line with prior work,47,48 households with children in our sample had a higher probability of being persistently food insecure or experiencing one or more food insecurity transitions. Before COVID-19, targeted policy efforts reduced child food insecurity to 13.6% in 2019, but economic shocks in 2020 caused a spike that was briefly alleviated to 12.5% by government waivers in existing safety net programs (e.g., expanding WIC options, maximizing SNAP allotments) and pandemic-era programs (e.g., Pandemic EBT, Child Tax Credit); however, the expiration of some waivers and of pandemic-era programs led to a sharp increase to 17.3% in 2022.45,46 Likewise we observed that food insecurity transitions and persistent food insecurity were more common among non-Hispanic Black and Hispanic respondents and respondents of low socioeconomic status.45,46,49,50 During COVID-19, these populations experienced higher unemployment 49–51 and higher case rates and deaths51 related to COVID-1952 compared to their non-Hispanic White and high socioeconomic counterparts, continuing pre-pandemic trends. Conversely, adults over the age of 65 in this analysis experienced fewer food insecurity transitions and were less likely to be persistently food insecure, a finding that has been reflected in other work, which may be due to the higher availability of food assistance programs for older rather than younger adults.53 Our work, in the context of a rich body of work examining U.S. food insecurity 54,55 highlights opportunities to build a more resilient food system and hunger relief program network through partnerships between researchers, hunger relief organizations, government organizations, and policymakers to identify populations in need of targeted policies and interventions.”

References

Bowen S, Elliott S, Hardison-Moody A. The structural roots of food insecurity: How racism is a fundamental cause of food insecurity. Sociol Compass. 2021;15(7):e12846. doi: https://doi.org/10.1111/soc4.12846

Tan S Bin, deSouza P, Raifman M. Structural Racism and COVID-19 in the USA: a County-Level Empirical Analysis. J Racial Ethn Health Disparities. 2022;9(1):236-246. doi: 10.1007/s40615-020-00948-8

Chapman LE, Hu J, Seidel S. COVID-19 Cases are Associated with Food Insufficiency in the United States During the COVID-19 Pandemic. J Hunger Environ Nutr. 2023;18(3):327-342. doi: 10.1080/19320248.2023.2202618

Norris K, Jilcott Pitts S, Reis H, Haynes-Maslow L. A Systematic Literature Review of Nutrition Interventions Implemented to Address Food Insecurity

---

## [Decision Letter · Decision Letter 1]

9 Mar 2025

Associations between early-pandemic food assistance use and subsequent food security trajectories among households in Washington State during the first three years of the COVID-19 pandemic

PONE-D-24-22286R1

Dear Dr. Buszkiewicz

We’re pleased to inform you that your manuscript has been judged scientifically suitable for publication and will be formally accepted for publication once it meets all outstanding technical requirements.

Kind regards,

Karen M Davison, PhD

Academic Editor

PLOS ONE

---

## [Editor Report · Acceptance letter]

PONE-D-24-22286R1

PLOS ONE

Dear Dr. Buszkiewicz,

I'm pleased to inform you that your manuscript has been deemed suitable for publication in PLOS ONE. Congratulations! Your manuscript is now being handed over to our production team.

Kind regards,

on behalf of

Dr. Karen M Davison

Academic Editor

PLOS ONE